# Decreased serum level of sphingosine-1-phosphate: a novel predictor of clinical severity in COVID-19

Giovanni Marfia[1,2,3,*,†] (ID), Stefania Navone[1,3,†], Laura Guarnaccia[1,4], Rolando Campanella[1], Michele Mondoni[5], Marco Locatelli[1,3,6], Alessandra Barassi[7], Laura Fontana[6], Fabrizio Palumbo[2], Emanuele Garzia[2,8], Giuseppe Ciniglio Appiani[2], Davide Chiumello[9], Monica Miozzo[6,10], Stefano Centanni[5,‡] & Laura Riboni[11,‡]

## Abstract

The severity of coronavirus disease 2019 (COVID-19) is a crucial problem in patient treatment and outcome. The aim of this study is to evaluate circulating level of sphingosine-1-phosphate (S1P) along with severity markers, in COVID-19 patients. One hundred eleven COVID-19 patients and forty-seven healthy subjects were included. The severity of COVID-19 was found significantly associated with anemia, lymphocytopenia, and significant increase of neutrophil-to-lymphocyte ratio, ferritin, fibrinogen, aminotransferases, lactate dehydrogenase (LDH), C-reactive protein (CRP), and D-dimer. Serum S1P level was inversely associated with COVID-19 severity, being significantly correlated with CRP, LDH, ferritin, and D-dimer. The decrease in S1P was strongly associated with the number of erythrocytes, the major source of plasma S1P, and both apolipoprotein M and albumin, the major transporters of blood S1P. Not last, S1P was found to be a relevant predictor of admission to an intensive care unit, and patient's outcome. Circulating S1P emerged as negative biomarker of severity/mortality of COVID-19 patients. Restoring abnormal S1P levels to a normal range may have the potential to be a therapeutic target in patients with COVID-19.

**Keywords** Coronavirus; COVID-19; intensive care unit; prognostic biomarker; sphingosine-1-phosphate
**Subject Categories** Biomarkers; Microbiology, Virology & Host Pathogen Interaction

See also: **H Rosen & MBA Oldstone** (January 2021)

## Introduction

In December 2019, a novel enveloped RNA betacoronavirus, named severe acute respiratory syndrome coronavirus 2 (SARS-CoV-2), rapidly spread from China throughout the globe, causing the pandemic respiratory disease COVID-19. Lombardy has been the most affected Italian region by the COVID-19 (Sotgiu *et al*, 2020) with > 175,000 cases, and 17,414 ascertained deaths (as of October 29, 2020, source: Italian Health Ministry).

The clinical spectrum of COVID-19 varies from asymptomatic/paucisymptomatic forms to critical conditions requiring support in an intensive care unit (ICU), and characterized by respiratory and multiple organ failure, systemic manifestations as septic shock, and even death (Guan *et al*, 2020; Gattinoni *et al*, 2020). The management of COVID-19 patients is constantly evolving, and so far, several medications are available, although without efficacy confirmation by clinical trials in most instances.

It is believed that the cytokine storm caused by systemic overproduction of pro-inflammatory cytokines, including interleukin-6 (IL-6), is a relevant cause of disease severity, progression, and death in COVID-19 patients (Song *et al*, 2020b). However, even though clinical features directly related to the rapid and intense inflammation, the lack of direct evidence makes unclear how the exuberant inflammatory process and organ failure is completed. Despite the importance of

---

1 Laboratory of Experimental Neurosurgery and Cell Therapy, Neurosurgery Unit, Fondazione IRCCS Ca' Granda Ospedale Maggiore Policlinico, Milan, Italy
2 Istituto di Medicina Aerospaziale "A. Mosso", Aeronautica Militare, Milan, Italy
3 Aldo Ravelli" Research Center, Milan, Italy
4 Department of Clinical Sciences and Community Health, Università degli Studi di Milano, Milan, Italy
5 Respiratory Unit, ASST Santi Paolo e Carlo, Department of Health Sciences, Università degli Studi di Milano, Milan, Italy
6 Department of Medical-Surgical Physiopathology and Transplantation, Università degli Studi di Milano, Milan, Italy
7 Laboratory of Clinical Biochemistry, ASST Santi Paolo e Carlo, Department of Health Sciences, Università degli Studi di Milano, Milan, Italy
8 Reproductive Medicine Unit, ASST Santi Paolo e Carlo, Università degli Studi di Milano, Milan, Italy
9 SC Anestesia e Rianimazione, ASST Santi Paolo e Carlo, Milan, Italy
10 Unit of Research Laboratories Coordination, Fondazione IRCCS Ca' Granda, Ospedale Maggiore Policlinico, Milan, Italy
11 Department of Medical Biotechnology and Translational Medicine, LITA-Segrate, Università degli Studi di Milano, Milan, Italy
*Corresponding author. Tel: +39 0255034268; Fax: +39 0255038821; E-mail: giovanni.marfia@policlinico.mi.it
†These authors contributed equally to this work as first authors
‡These authors contributed equally to this work as senior authors

---

the older age and underlying comorbidities as prognostic factors associated with severity/death of COVID-19 patients, the evolution of SARS-CoV-2 infection is often very heterogeneous and unpredictable, and robust prognostic biomarkers, and potential therapeutic targets are still lacking. Thus, the actual, rapidly expanding knowledge on COVID-19 highlights the need for a better understanding of its pathophysiology, and for developing novel biomarkers with efficacious prognostic value, and potential therapeutic applications.

Among multiple mediators of inflammation, sphingosine-1-phosphate (S1P) emerged as a key signal (Obinata & Hla, 2019), involved in the regulation of multiple pathophysiological processes, including vascular physiology, and immunity (Yanagida & Hla, 2017; Bryan & Del Poeta, 2018). This bioactive sphingoid is produced intracellularly by sphingosine kinases, and, after extracellular release, it exerts pleiotropic effects through binding to specific G protein-coupled receptors (S1P1–5; Liu *et al*, 2012). In physiological conditions, S1P is present at high concentrations in blood, by the contribution of erythrocytes (Hänel *et al*, 2007), endothelial cells (Venkataraman *et al*, 2008), and platelets (Yatomi *et al*, 2000). To the opposite, its levels in other tissues are low, creating a vascular S1P gradient, crucial for S1P to exert its regulatory roles (Yanagida & Hla, 2017). In the systemic circulation, about 60% of plasma S1P is carried by high-density lipoproteins (HDL), bound to apolipoprotein M (apoM), and about 30% by albumin, and the biological properties of S1P attached to ApoM and albumin are different (Kurano & Yatomi, 2018). On these premises and taking into account that S1P is involved in viral infections (Wolf *et al*, 2019), and in sepsis (Winkler *et al*, 2019), the objective of this study was to determine the serum levels of S1P, and its transporters apoM and albumin, to evaluate their clinical importance as prognostic/predictive biomarkers in COVID-19.

# Results

### Characteristics of the study population

Demographic and clinical characteristics of the two groups are shown in Table EV1. HLT and COV groups were very similar for sex and age, as well as for hypertension or diabetes incidence.

The HLT population presented all the biochemical parameters within the normal range, whereas COV patients exhibited a pattern of hematological and biochemical abnormalities. In particular, we found significant differences in RBC, HGB, HCT, RDW, WBC, neutrophils, lymphocytes, and monocytes values (Table EV2). Moreover, the calculated neutrophils-to-lymphocytes ratio (NLR) accounted for 1.78 (IQR: 1.47–2.09) in HLT subjects and 7.05 (IQR: 5.34–8.85) in COV patients, with a strong statistically significant difference ($P < 0.0001$) between the two groups.

Among the biochemical analytes, we found multiple statistically significant differences between COV and HLT (Table EV3), including significant decrease in total proteins, albumin, total cholesterol, HDL-C, 25-OH vitamin, and significant increase in fibrinogen, urea, ferritin, GGT, AST, ALT, CRP, LDH, NT-proBNP, and IL-6. Total bilirubin, albeit within the normal range, was significant lower in COV patients than in HLT. Furthermore, the coagulation profile was altered in COV patients with a significant increased level of D-dimer and fibrinogen. To the contrary, coagulation parameters PT and

aPTT, and electrolytes, including calcium, sodium, potassium, and chloride were similar in HLT and COV cohorts.

ELISA assays provided a mean value of serum S1P concentration in the HLT group of 0.87 μM, which is very similar to that reported by a recent study on 174 healthy blood donors, and obtained by LC/MS/MS (Daum *et al*, 2020). The measurement of serum levels of S1P and its transporter apoM revealed a highly significant decrease in both molecules in COV patients, compared to HLT subjects (Fig 1A and B). In particular, the median S1P values were 0.87 and 0.69 μM (Fig 1A), and the median apoM values were 39.0 and 24.3 μg/ml (Fig 1B), in HLT and COV, respectively. These results suggest a systemic involvement of S1P/apoM complex in COVID-19. However, the Pearson correlation between S1P and apoM gave no statistical significance, even though a modest positive trend was present (Fig 1C). Interestingly, serum S1P levels significantly correlated with RBC, HGB, and HCT values (Fig 1D–F).

The Pearson correlation demonstrated a significant correlation between S1P and both HDL-C (Fig 2A), and albumin (Fig 2B). In parallel, significant correlations were found between apoM and HDL-C (Fig 2C) and albumin (Fig 2D). To the contrary, no significant correlation was found between PLT and S1P, as well as PLT and apoM.

### Identification of disease's severity markers

To investigate the role of S1P and apoM as circulating biomarkers in COVID-19 infection severity, we stratified patients based on whether or not they needed ICU admission, categorizing our COV patient cohort into ICU and noICU groups. As shown in Table EV4, both groups were homogeneous in terms of age and BMI. As expected, ICU patients showed a significant increase in mortality rate, even if their frequency of hypertension and diabetes was significantly lower than that of noICU. Furthermore, the comparison between blood and biochemical parameters revealed that ICU patients, compared to noICU, have a significant decrease in RBC, HGB, HCT, and lymphocytes, concomitant to a significant increase in WBC, neutrophils, NLR, urea, AST, ALT, LDH, CRP, and D-dimer (Table EV5). Intriguingly, patients admitted to ICU had significantly lower values of S1P, apoM, and of albumin and HDL-C, compared to the noICU group (Fig 3A–D), suggesting the potential role of S1P and its blood transporters apoM, albumin, and HDL-C, as circulating biomarkers of disease severity.

To determine the possible relation between serum levels of S1P and apoM and the severity of infection in COVID-19 patients, we used PSI, length of hospitalization, and NLR as parameters. Interestingly, a significant negative correlation was found between S1P and PSI, days from hospital admittance, and NLR (Fig 4A–C). Similarly, significant negative correlations were found between apoM, PSI, and NLR (Fig 4D and F), but not between apoM and days from hospital admission (Fig 4E).

In univariable logistic regression analysis, PSI, S1P, apoM, RBC, HGB, HCT, WBC, NLR, CRP, and albumin were significantly associated with ICU admission. Results of multivariable regression analysis showed that NLR and S1P are independent predictors for ICU admission. Noteworthy, S1P, among the considered parameters, is the most important risk factor for ICU admission for COVID-19 patient (OR: 39.45, [95% CI: 1.51–1031.60]; $P = 0.027$; Table EV6). Further, to estimate the potential prognostic role of S1P, COV

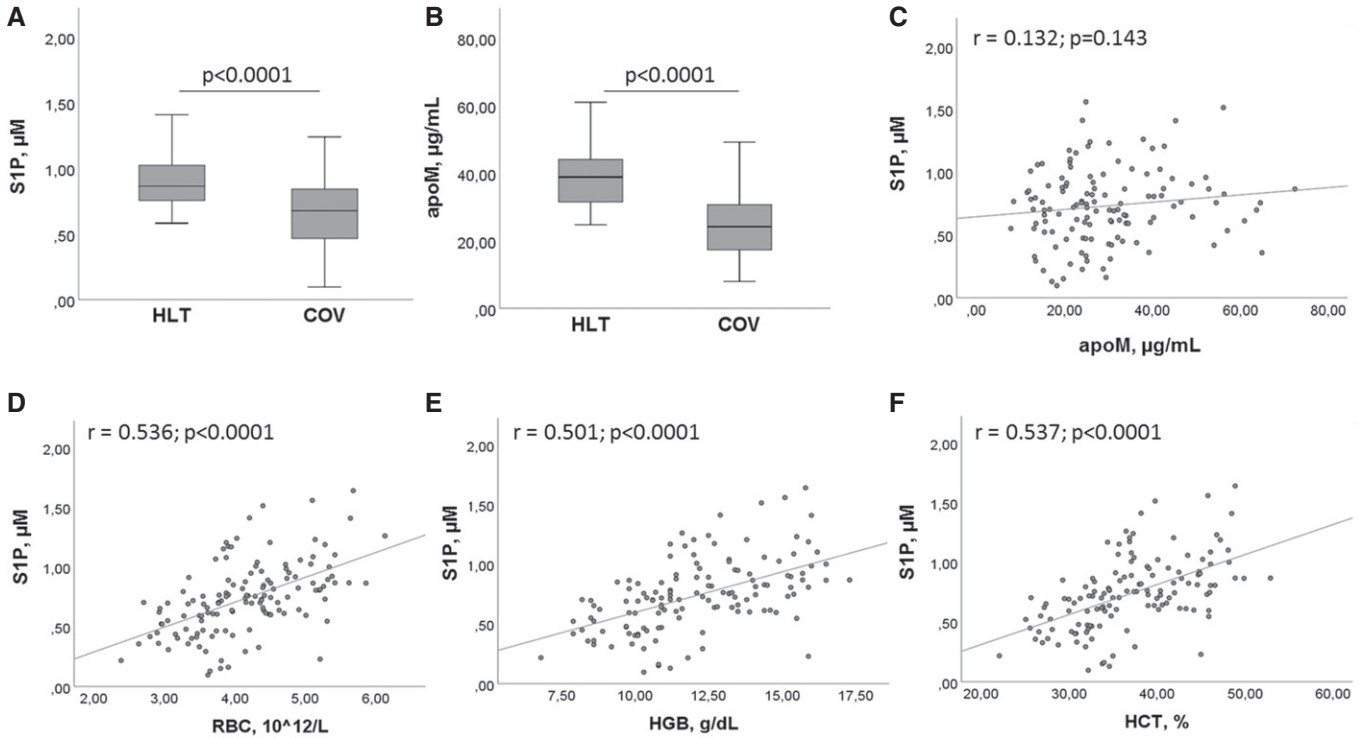

**Figure 1. Serum levels of S1P and apoM in HLT and COV patients.**

A, B  Serum levels of S1P (A) and apoM (B) in HLT (n = 47) and COV (n = 111) patients. The box plots represent the interquartile range with median value (central line); the whiskers represent the measured range of HLT and COV. Each measurement was run in triplicate, and performed at least twice. Two-tailed Student's *t*-test was used for statistical analysis. *P* < 0.0001 is reported.

C–F  Pearson correlation between S1P and apoM (C), RBC number (D), HGB concentration (E), HCT value (F). Scatter plots, together with the fitted regression line, are shown. Pearson correlation was performed for statistical analysis. Exact *P* values or *P* < 0.0001 are reported.

patients were stratified by the cutoff value of S1P 0.60 μM, according to Youden index. Cox regression analysis revealed that S1P is a significant predictor of ICU admission, and in-hospital mortality (Fig 5A and B). Of relevance, the mortality rate in COV patients with S1P < 0.60 μM (13.6%) was found significantly higher (*P* < 0.05) than that with S1P ≥ 0.60 μM (4.5%) In addition and interestingly, patients admitted to ICU with low serum S1P showed a meaningful rise of mortality rate up to 33%.

## Discussion

The major findings of this study are that serum S1P levels are significantly lower in COVID-19 patients than HLT and predict both ICU admission and in-hospital mortality. A very recent lipidomic study, published while our paper was in preparation, reported that S1P is significantly reduced in plasma samples of COVID-19, without indicating molar concentrations, which prevents comparison across independent studies, and without addressing to disease severity and patient outcome (Song *et al*, 2020a). To our knowledge, this is the first study providing evidence for serum S1P association with COVID-19 severity, and suggesting S1P as a novel circulating biomarker of COVID-19 severity and morbidity.

Previous studies demonstrated that the cytokine storm plays a key role in COVID-19, and the increase of pro-inflammatory

cytokines is associated with disease severity (Song *et al*, 2020b). Of note, S1P was implicated in both upstream and downstream cytokine production and increased interstitial levels of S1P at the inflammatory sites induce the expression of pro-inflammatory cytokines (Obinata & Hla, 2019), suggesting that local, interstitial S1P may concur to the cytokine storm of COVID-19. Taking into consideration the relevance of the cytokine storm in COVID-19, we here discuss our findings on serum S1P in COVID-19 patients through two main sections: (i) the possible mechanisms underlying decreased level in serum S1P, including cellular contributors of circulating S1P, and its blood transporters; (ii) the functional consequences and clinical implications of low circulating S1P. To facilitate the reader, Fig 6 provides an overview of the proposed mechanisms underlying S1P involvement in COVID-19.

### Mechanisms underlying the reduction of serum S1P

Due to their unique S1P metabolism and processing, erythrocytes are a major source of S1P in blood plasma, their count positively associates with plasma S1P levels (Hänel *et al*, 2007). We found that COVID-19 patients have a significantly reduced RBC count, and that this count, together with total hemoglobin and hematocrit values were directly correlated with serum S1P. These findings, and the reports on low blood S1P in anemic patients (Ohkawa *et al*, 2008), strongly suggest that RBC decrease may

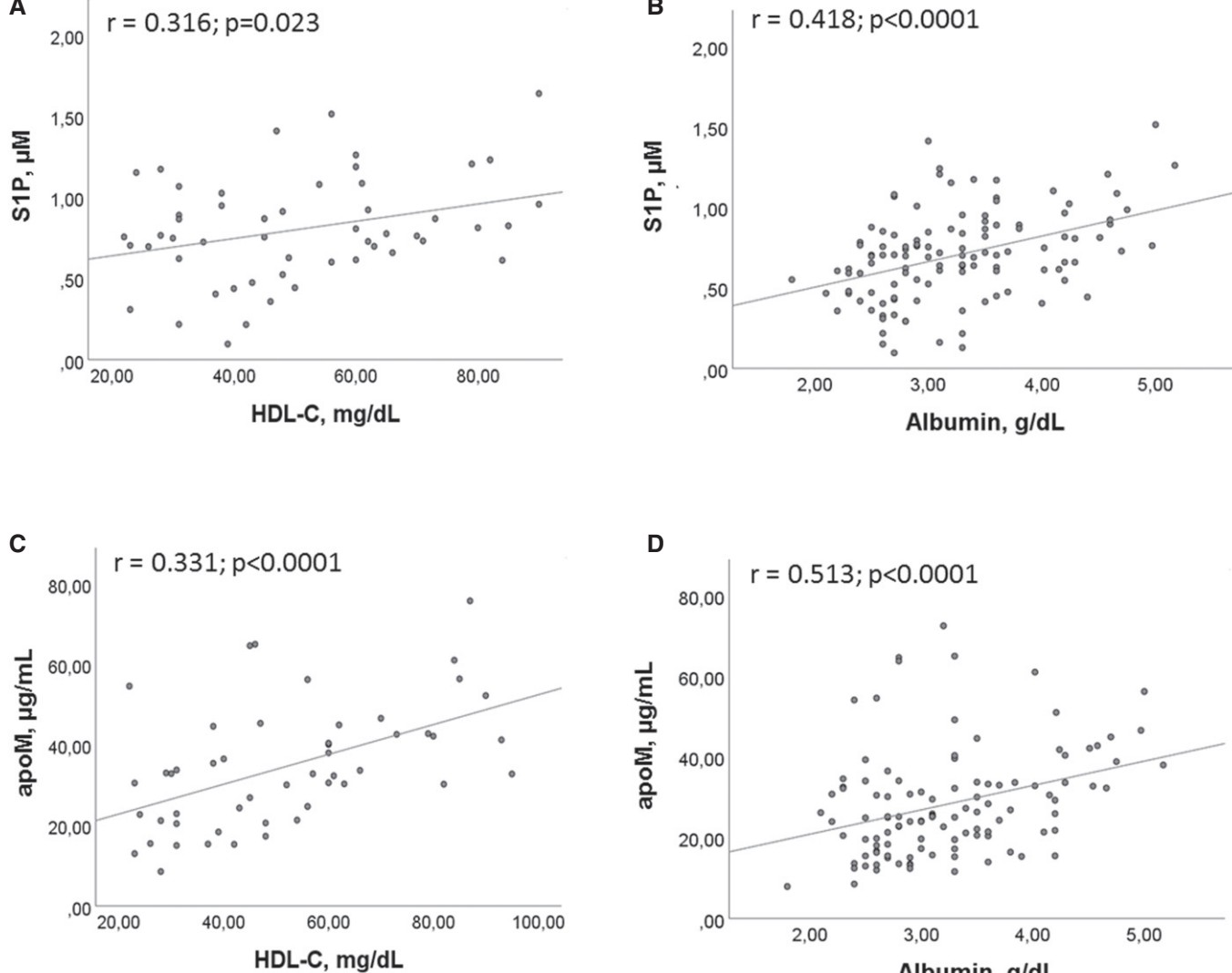

**Figure 2. Correlations between S1P/apoM and their transporters.**

A, B  Correlations between S1P and HDL-C (A), and albumin (B).

C, D  Correlations between apoM and HDL-C (C) and albumin (D).

Data information: Scatter plots and fitted regression line are shown in each figure. Each S1P and apoM measurement was run in triplicate, and performed at least twice in independent assays. Pearson correlation was performed for statistical analysis. Exact *P* values or P < 0.0001 are reported.

contribute to low serum S1P in COVID-19 patients. Although the mechanism-underlying anemia in COVID-19 is unknown, a possible explanation resides in the finding that pro-inflammatory cytokines inhibit erythropoietin-induced erythropoiesis (Pierce & Larson, 2005). This process was associated with RDW increase (Pierce *et al*, 2005), which we found in our COVID-19 cohort. Furthermore, as a very recent preprint notice reported a significant sphingolipid decrease in RBC from COVID-19 patients (Thomas *et al*, 2020), we cannot exclude that a decrease in S1P synthesis/export by erythrocytes might also occur. In addition to erythrocytes, the vascular endothelium efficiently contributes to plasma S1P levels through the plasma membrane S1P-transporter Spns2 (Nagahashi *et al*, 2013). As pro-inflammatory cytokines down-regulate Spns2 in endothelial cells (Jeya Paul *et al*, 2020),

and thus their S1P export, it is possible that the cytokine storm in COVID-19 patients contributes to an endothelial-induced S1P drop. In agreement, patients with sepsis have decreased serum S1P levels (Winkler *et al*, 2019).

A further source of blood S1P are platelets, which efficiently store S1P, and mainly release it into the circulation during clotting, leading to higher S1P levels in serum than plasma (Yatomi *et al*, 2000). Our COVID-19 cohort had platelet numbers and parameters in the normal range, most probably excluding platelets as responsible for their low serum S1P.

Besides possible alterations of S1P release into blood, our data reveal that decreased levels of apoM and albumin, the two major S1P transport proteins, are likely involved in circulating S1P reduction. Our study unraveled three principal findings regarding S1P

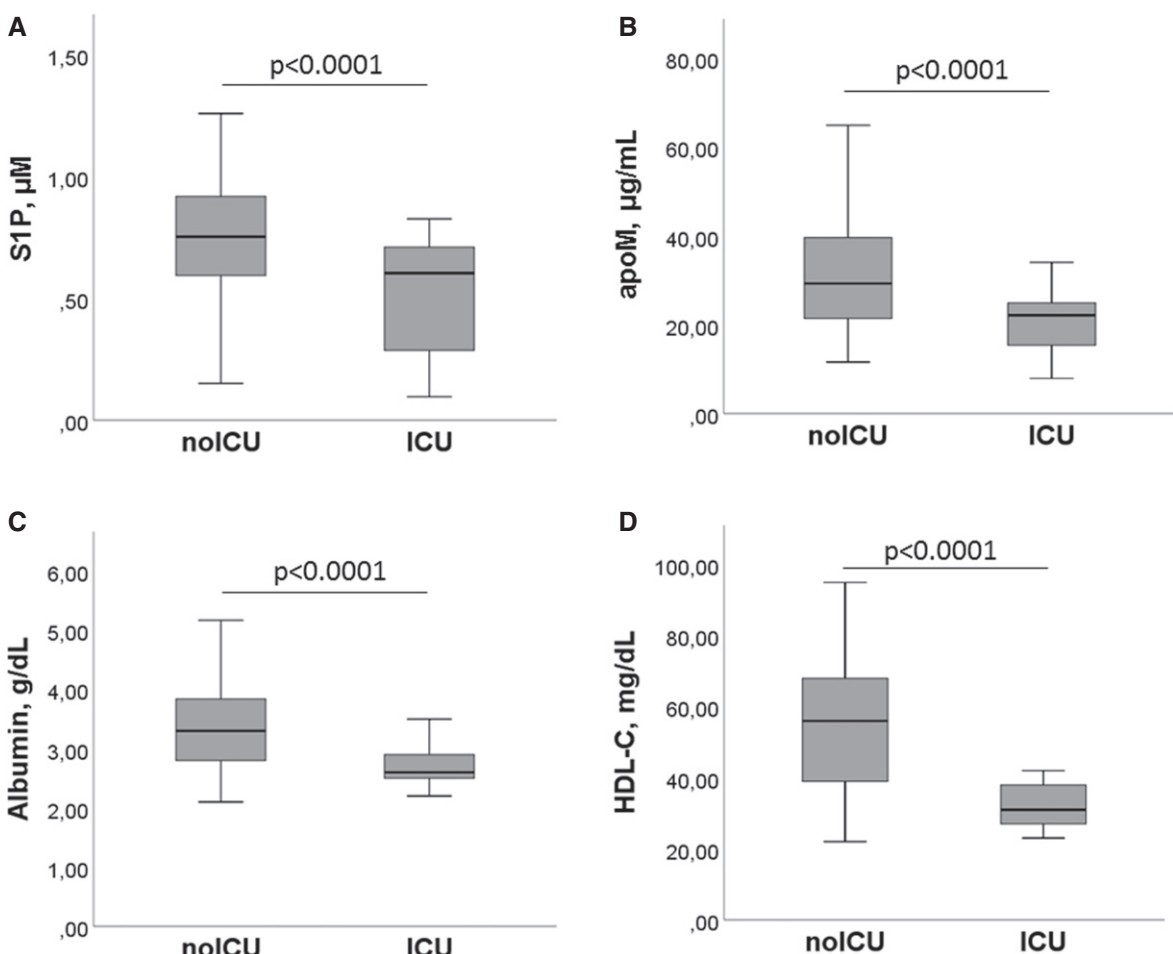

**Figure 3.  Serum levels of S1P and its blood transporters in noICU and ICU patients.**

A–D   The serum concentrations of S1P (A), apoM (B), albumin (C), and HDL-C (D) in noICU ($n = 89$) and ICU ($n = 22$) patients are shown. The box plots represent the interquartile range with median (central line); the whiskers represent the measured range of noICU and ICU patients. Each S1P and apoM measurement was run in triplicate, and performed at least twice in independent assays. Two-tailed Student's $t$-test was used for statistical analysis. $P < 0.0001$ is reported.

carriers in COVID-19 patients. First, we found that serum apoM and albumin were both significantly lower in COVID-19 patients compared to HLT, and particularly in ICU than noICU patients. As these proteins, besides acting as S1P vehicle, are essential for S1P release from erythrocytes, and modulate blood S1P concentration (Christensen *et al*, 2017), their decrease most probably concurs to the drop of circulating S1P in COVID-19 patients. Of note, both apoM and albumin are considered negative acute-phase proteins (Feingold *et al*, 2008), as their levels profoundly decrease during inflammatory conditions. Here, we show that concomitant to albumin and apoM reduction, the serum levels of CRP and ferritin, widely recognized as positive acute-phase reactants that rise dramatically as part of the inflammatory response mediated by increased cytokines such as IL-6 (Streetz *et al*, 2001), significantly increased in COVID-19 patients. Thus, it appears reasonable that apoM and albumin drop as part of the inflammatory response mediated by the cytokine storm in COVID-19 patients.

A second interesting observation of our study is that serum values of S1P significantly correlate with HDL-C, and are lower in ICU than in noICU patients, indicating that serum S1P levels are influenced by

HDL, and are related to COVID-19 severity. This is in agreement with reports showing that: (i) plasma S1P levels positively correlate with plasma levels of HDL-C (Zhang *et al*, 2005), (ii) severe forms of sepsis result in severely decreased plasma levels of HDL-C, (iii) HDL-S1P relates to sepsis severity, and contributes most to the severe drop of total plasma S1P found in septic shock patients (Winkler *et al*, 2019). Third, the serum levels of S1P, although related to HDL-C, did not significantly correlate with serum apoM levels, indicating that the low apoM levels in patients are probably not directly linked to their decreased S1P level. The reason of this discrepancy might be that S1P is linked also to albumin and other HDL–apolipoproteins, and that apoM can bind lipids other than S1P.

### Pathological consequences and clinical implications of reduced serum S1P

Pathological key features of COVID-19 are complex, and its pathogenesis remains unclear.

Here, we report current evidence on the multiple effects of blood S1P, discussing them in the context of reduced serum S1P, together

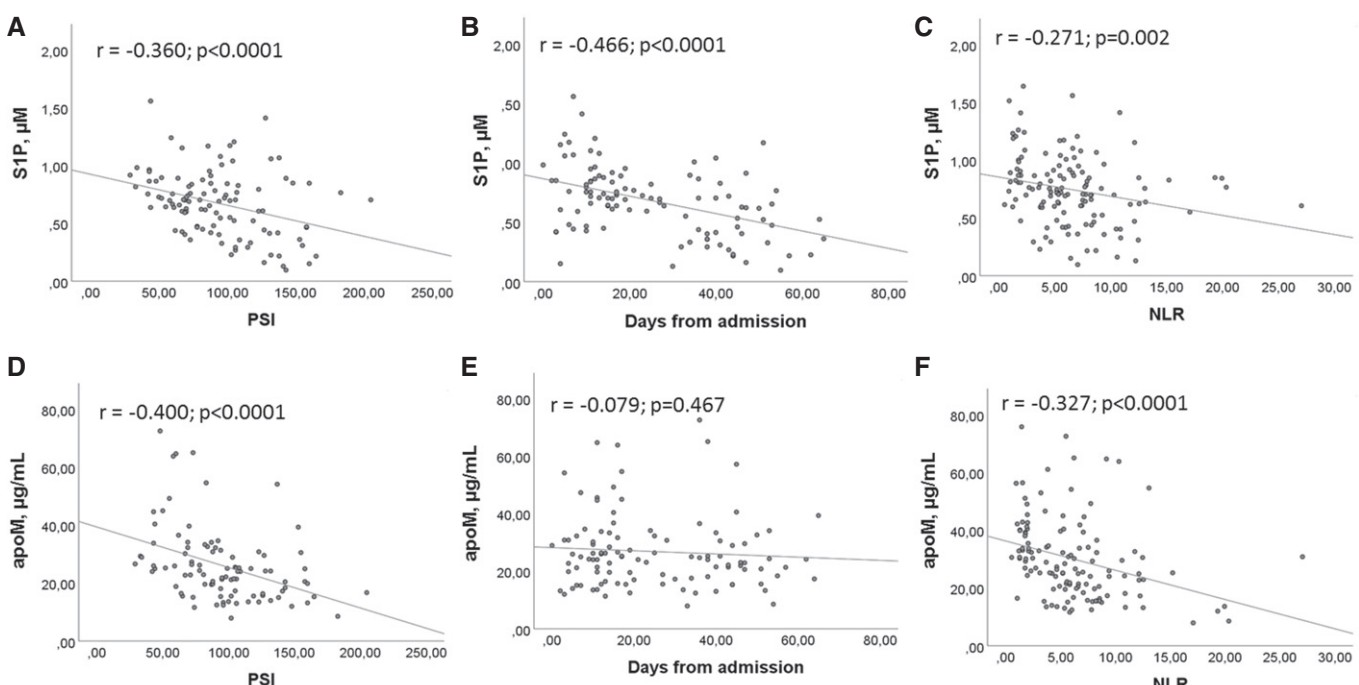

**Figure 4. Correlation between S1P/apoM and COVID-19 infection severity.**

A–F  Pearson correlation between S1P (A–C) or apoM (D–F) and PSI (A, D), days from admission (B, E), and NLR (C, F) in COV (*n* = 111). Scatter plots and fitted regression line are shown in each figure. Exact *P* values or *P* < 0.0001 are reported.

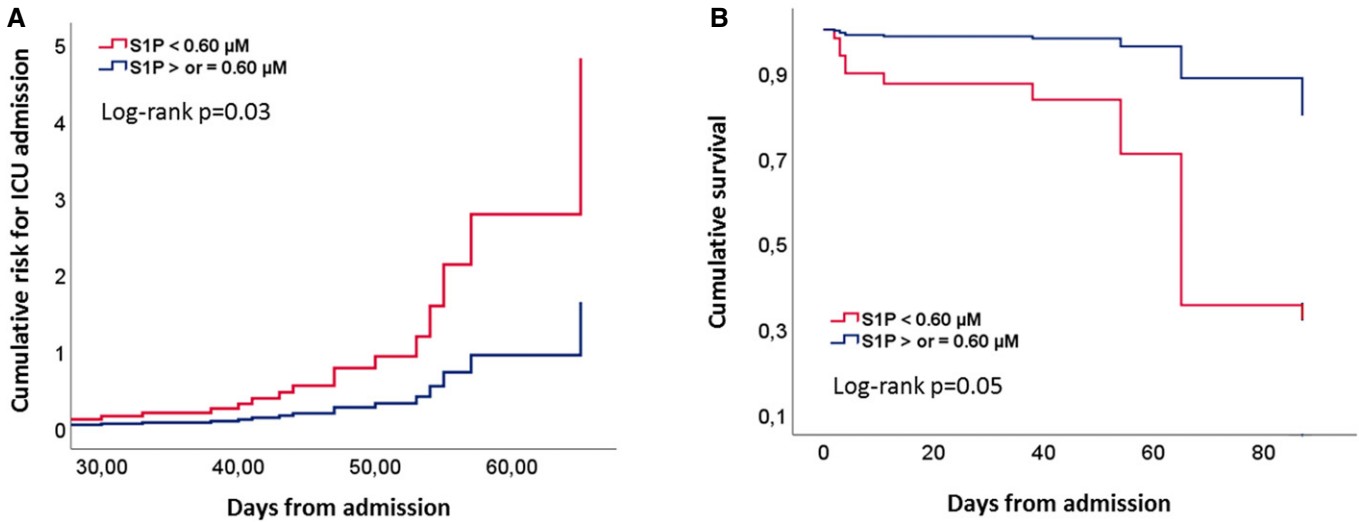

**Figure 5. Prognostic value of S1P in COV patients.**

A, B  Cumulative risk for ICU admission (A) and cumulative survival (B) in COV patients (*n* = 111), grouped for cutoff value of S1P serum level of 0.60 µM. Statistical analysis was performed by Cox regression. Exact *P* values are reported.

with its clinical implication in COVID-19. In agreement with previous findings (Guan *et al*, 2020), we found that COVID-19 patients present lymphopenia. The mechanisms underlying lymphopenia remain unknown, and we hypothesize that the decrease in S1P levels might participate. Indeed, the S1P gradient between blood and tissues is essential for lymphocyte egress from lymphoid organs, which occurs

via S1P1 (Baeyens & Schwab, 2020). In agreement, FTY720, a functional antagonist of the S1P/S1P1 signaling pathway approved as immunosuppressant, produces peripheral lymphopenia (Pelletier & Hafler, 2012). Of interest, the S1P1-mediated trafficking and egress of lymphocytes are limited to albumin-S1P, and not shared by apoM-S1P (Baeyens & Schwab, 2020). On these bases, our findings on the

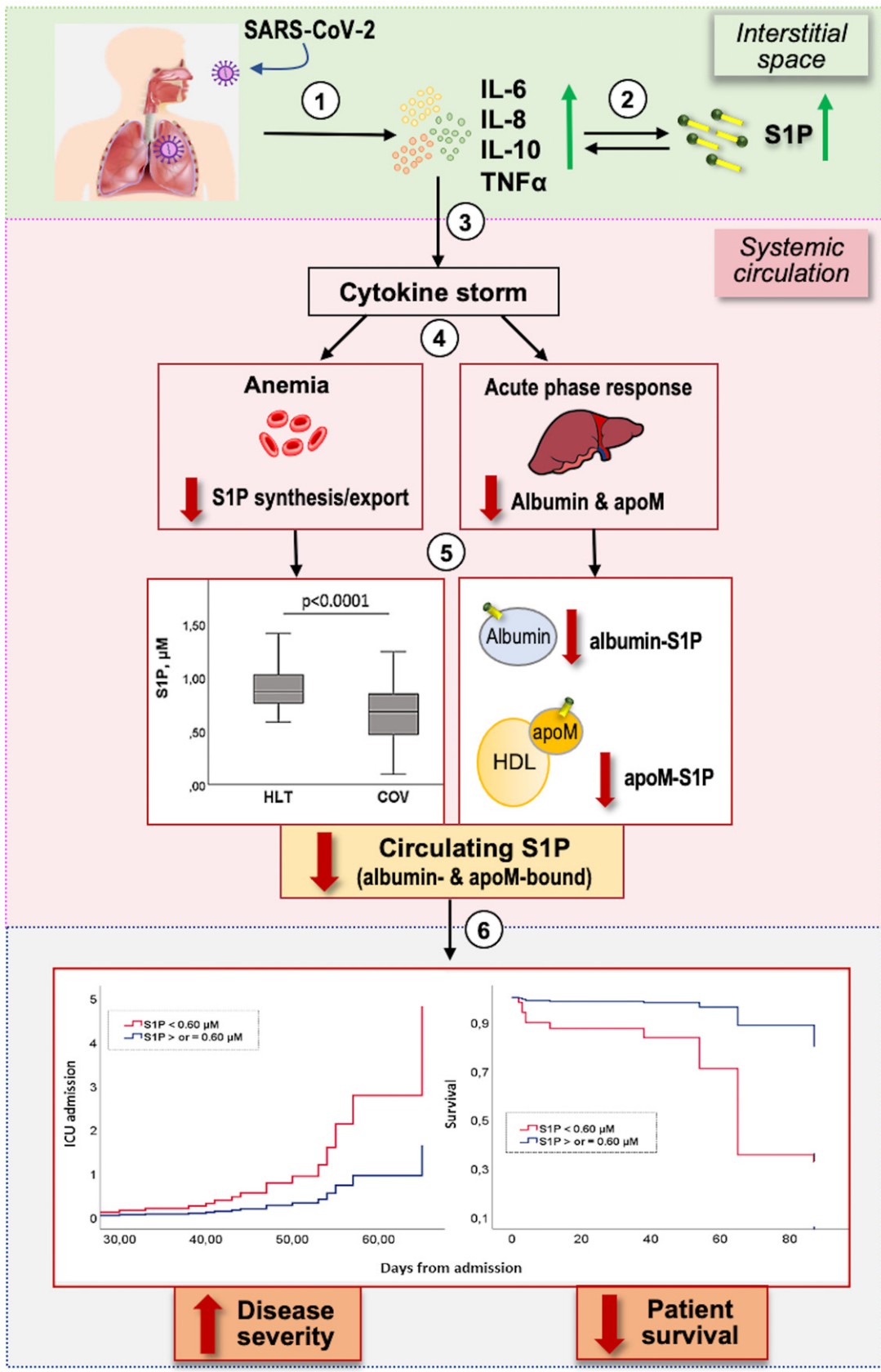

**Figure 6.**

**Figure 6. Overview of the proposed mechanisms underlying S1P involvement in COVID-19 pathophysiology and severity.**

After SARS-CoV-2 infection, a local inflammation occurs, with increased pro-inflammatory cytokines (1). This promotes an interstitial increase of S1P, which in turn potentiates cytokine secretion by different cells (2). The exuberant local cytokine levels result in a systemic cytokine storm (3). This gives on to different alterations (4), including anemia (with impaired S1P synthesis/export) and acute-phase response in the liver (with decrease in the negative acute-phase proteins albumin and apoM, which act as S1P transporters). These alterations lead to a progressive drop of circulating S1P, with decrease in both apoM/S1P and albumin/S1P (5). The reduction of S1P in the systemic circulation correlates with COVID-19 severity and patient outcome (6).

reduction of both S1P and albumin in serum from COVID-19 patients might have implications on COVID-19 lymphopenia, and related immunosuppressive features.

A further relevant physiological role of circulating S1P consists in its protective and anti-inflammatory actions on endothelial cells (Christoffersen et al, 2011). ApoM/S1P exerts more potent activities on endothelial cells than albumin/S1P, and particularly in limiting endothelial inflammation, and maintaining the endothelial barrier (Christoffersen et al, 2011). As described above, free interstitial S1P increases at inflammation sites, where, opposite to its plasma anti-inflammatory effects, it is involved in the propagation of inflammation. Considering the scavenger actions of apoM/HDL (Kurano & Yatomi, 2018), which are involved in inflammation resolution by neutralizing S1P excess at inflammation sites, the deficiency of apoM might contribute to exacerbate inflammation in COVID-19. Overall, despite we cannot exclude that the decreased levels of S1P could be an epiphenomenon of host tissue damage, based on its protective properties on endothelial barrier and anti-inflammatory properties of apoM/S1P, the decrease in S1P and apoM might explain the disturbed vascular barrier function, and organ injuries of COVID-19. In agreement, it was previously reported that, during septic challenge, the plasma levels of S1P drop to very low levels, the liver synthesis of apoM decreases severely and the plasma levels of apoM are reduced. Thus, the decrease in circulating S1P may contribute to the clinical severity of pathologies involving endothelial barrier dysfunction as sepsis (Winkler et al, 2019), peripheral artery disease (Soltau et al, 2016), and liver cirrhosis (Becker et al, 2017).

Further interesting findings of our study refer to the relation between S1P (and apoM) and the established inflammation marker NLR, as well as the clinical prediction tool PSI, both found correlated with COVID-19 severity (Ciccullo et al, 2020; Satici et al, 2020). We report here that both NLR and PSI were not only significantly higher in ICU than noICU group, but also correlated with serum S1P and apoM, and that S1P negatively correlates with the length of hospitalization. Furthermore, when COVID-19 patients were stratified using a cutoff for S1P, the patients with low serum level displayed significant increased probability of ICU admission and mortality rate. Overall, these correlations strongly suggest that circulating S1P levels may be clinically used as negative biomarkers to predict severity/mortality of COVID-19 patients.

Finally, our study suggests that restoring low S1P and its transporters to healthy range may be a therapeutic target for reducing COVID-19 severity, and death. Contrasting results were found on Fingolimod, an FDA-approved functional antagonist of S1P. Indeed, accordingly previous data reporting a potential role for Fingolimod to blunt cytokine storm induced by (Oldstone & Rosen, 2014) H1N1 and influenza virus (Walsh et al, 2011) infection, a clinical trial for Fingolimod administration in COVID-19 patients has been initiated and then discontinued in some patients for marked lymphopenia (www.clinicaltrial.gov NCT04280588). Furthermore, glucocorticoids, widely used in the treatment of COVID-19 with contra

dictory effects (Ledford, 2020), target S1P signaling, by inhibiting S1P-induced cytokine secretion (Che et al, 2014). Our findings strongly suggest that S1P circulating levels should be measured in COVID-19 patients before administration of any S1P-modulating drug, including glucocorticoids, and that therapeutic modalities (local/systemic) should be evaluated before treatment.

Our study has some limitations. First, it is a single-center study, with a relatively small sample size. This could have reduced the power of the study, but not the novel finding on the reduction of S1P and apoM in the serum of COVID-19 patients. Further multi-center studies with a larger patient cohort are needed to in-depth evaluate the here-described correlation of S1P levels with circulating parameters and disease severity of COVID-19. Second, as plasma S1P levels should be measured in samples collected under specific conditions, and platelets may leak S1P during blood storage, we used serum to measure S1P, with the consequent influence of platelet-derived S1P. Further studies are needed to elucidate the association between S1P and COVID-19 by using plasma, and by separating S1P bound to HDL from S1P bound to albumin. Third, we performed the evaluation of serum S1P by an ELISA assay, which may have pitfalls. Although our data were validated by an enzymatic method, and S1P levels in the control group were comparable with those reported by HPLC/MS/MS, an accurate quantification of serum S1P levels through this technique will strengthen future findings. Finally, we were unable to collect blood samples during hospitalization to measure S1P at different time points in the same patients. Answering these questions in future studies will help to determine whether S1P and/or apoM could be markers to facilitate the management of COVID-19 in clinical practice, and potential helpful targets for COVID-19 treatments.

In conclusion, the current study establishes the decrease in serum S1P and apoM as novel circulating biomarkers associated with COVID-19 severity and morbidity. We speculate that S1P may play a role in endothelial barrier dysfunction, altered immune response, and persistent excessive inflammation in COVID-19 patients. Understanding the mechanisms leading to decreased serum S1P in COVID-19, and its multi-systemic effects should be the focus of future work, to advance our knowledge of this disease. Finally, the present investigation suggests that restoring abnormal S1P levels to a normal range, and balancing its binding to albumin and apoM to healthy conditions, may have the potential to be a therapeutic target for reducing the risk of disease progression and death, and, not last, to mount an effective immune response after vaccination in patients with COVID-19.

## Materials and Methods

### Study population and clinical variables

This prospective, case–control study was approved by the Ethic Committee of Ospedale San Paolo in Milan, Lombardy, Italy (COST

Action n.2020/ST/057). Patients with confirmed positivity to SARS-CoV-2 by molecular test of nasopharyngeal swabs (COV, $n = 111$), were enrolled consecutively during hospitalization in March–May 2020 at Ospedale San Paolo, outbreak epicenter of the Italian pandemic cluster. Also, healthy subjects (HLT, $n = 47$) were tested. Informed consent was obtained from all human subjects, and the experiments were conformed to the principles set out in the WMA Declaration of Helsinki and the Department of Health and Human Services Belmont Report. Blood sample collection was performed the day of the hospital admission. During hospitalization, a subgroup of these patients with severe symptoms were admitted to the Intensity Care Unit (ICU). In addition, Pneumonia Severity Index (PSI) was calculated at time of admission. Electronic data on demographics, medical history and comorbidities, illness onset and symptoms, vital signs, and baseline plasma/serum-based analytes were recorded for all admitted patients.

## Measurement of serum S1P

For serum S1P and apoM measurements, an aliquot (5 ml) of whole blood was collected in covered vacutainer without anti-coagulant. Tubes were left at room temperature for 30–45 min, and then were centrifuged (2,000 $g$, 15 min). The obtained serum, inspected to assess limpidity, was carefully transferred into clean tubes, aliquoted, and stored at −80°C until use, without repeated freeze–thaw cycles. The S1P levels were measured in duplicate, in at least two different serum aliquots from each patient, by enzyme-linked immunosorbent assay (ELISA) assay kit (Echelon Biosciences, Salt Lake City, USA), according to manufacturer's instruction. The standard S1P curve ranged from 0.03 to 4 μM, and a semi-log analysis was used to interpole unknown samples. To assess ELISA specificity, different concentrations of exogenous S1P were added to HLT and COV sera. A sigmoidal curve, very similar in HLT and COV samples, was obtained, excluding serum interferences. The linearity of the serum measurement was initially checked on 4 samples at two different dilutions (1:5 and 1:10). Then, all samples were diluted 1:10 in delipidized serum, and known low and high S1P standards were run in parallel with experimental samples, as controls. Each S1P measurement was run in triplicate, and performed at least twice in independent assays. To validate ELISA assays, five serum samples from both controls and COVID-19 patients were submitted to double partitioning (first in alkaline, and then in acidic conditions), and S1P concentration in the final acidic organic phase was measured by an enzymatic assay as previously reported (Edsall et al, 2000), with minor modifications (Abdel Hadi et al, 2018). The results demonstrated very similar values with the two methods, the difference of the coefficient of variation (the ratio of the standard deviation over the mean of the measurements) being < 14% in both groups.

## Measurement of serum apoM

ApoM levels were measured in serum (diluted 1:20,000) by the Human apoM ELISA[PRO] kit (Mabtech, Inc., Cincinnati, USA). Samples with known low and high concentrations of apoM were tested together with the study participant samples. The range of apoM standard curve was 0.03–20 ng/ml, and the 4-parameter curve fit was used to interpole sample measurements. Each apoM

### The paper explained

#### Problem
Since December 2019, COVID-19 has widely spread throughout the world, causing more than one million of deaths. Although clinical characteristics of COVID-19 patients have been widely reported, significative disease-associated biomarkers remain unknown. We evaluated the potential of circulating sphingosine-1-phosphate (S1P) as a prognostic and predictive biomarker in COVID-19.

#### Results
We report demographic, clinical, and laboratory findings of 111 patients with COVID-19, compared to 47 healthy subjects. Several blood parameters, including erythrocyte and lymphocyte number, neutrophil-to-lymphocyte ratio, and biochemical variables encompassing albumin, ferritin, D-dimer, and fibrinogen were found different in COVID-19. The novel, major finding of this work was the significant decrease of serum S1P level in COVID-19 patients, which was significantly related to the decrease of erythrocytes, the major cellular source of circulating S1P, as well as of the two key S1P transporters apoM and albumin. Of relevance, the serum levels of S1P, RBCs, apoM, and albumin exhibited the lowest values in patients admitted to intensive care unit (ICU). Multivariate logistic regression analyses revealed that S1P was the only parameter significantly associated with ICU admission, as well as the strongest predictor for both ICU admission and mortality risk.

#### Impact
The results of this study establish S1P as a novel circulating biomarker negatively associated with COVID-19 severity and morbidity, and shed light on the pathophysiology of the disease. Interventions to restore S1P abnormal levels should be considered as a potential therapeutic strategy for reducing the hazard of disease progression and death, and for mounting an effective immune response in patients with COVID-19.

measurement was run in triplicate, and performed at least twice in independent assays.

## Statistical analyses

The calculation of power size was performed considering a power of 80%, a type I error rate of 5%, and an effect size of 50%. The continuous variables were expressed as median values and interquartile range (IQR). Discrete variables were reported as counts or percentages. Blood parameters were tested for normality using the Kolmogorov–Smirnov and Shapiro–Wilk tests, and when normally distributed, the two conditions were compared by the two-tailed Student's $t$-test. The Pearson correlation test was used to assess the univariate association between variables.

The primary clinical endpoint of our study was ICU admission. The association of baseline characteristics and clinical findings with ICU admission was initially evaluated using univariable logistic regression. Variables with $P < 0.05$ were considered as potential risk factors and included in the multivariable logistic regression analysis with the backward stepwise method, in order to explore variables that were independently associated with ICU admission. The cutoff was calculated according to the maximum value of Youden index, and the hazard ratio for ICU admission and mortality was calculated using the Cox proportional hazard model. Statistical analysis of data was made

using IBM SPSS Statistics 26.0 software. Data acquisition was performed blindly. The tests were all two-sided and were considered statistically significant when $P < 0.05$. In all cases, we report exact $P$ value of the analyzed data, except when $P < 0.0001$, reported as such, as it is widely recognized as highly statistically significant.

## Data availability

The data used to support the findings of this study are available from the corresponding author upon request.

**Expanded View** for this article is available online.

### Acknowledgments

This work was supported by Fondazione IRCCS Ca' Granda Ospedale Maggiore Policlinico (RC2020 to GM). The present study was born thanks to the collaboration between civil and military health system, such as the Italian Air Force Medical Corps, which actively participated in management of COVID-19 outbreak during the most critical phases of the emergency in Lombardy and throughout Italy. The work was also arranged thank to the Framework Agreement between the IRCCS Ca' Granda Ospedale Maggiore Policlinico, Italian Air Force, and University of Milan. We apologize for not mentioning all the relevant articles pertinent to our study, due to limit of references. We thank all patients and their family members and all the frontline health workers for their efforts in the diagnosis and treatment of patients during COVID-19 pandemic.

### Author contributions

GM, SN, LG, and LR had the idea for and designed the study. GM, SN, LG, MMo, RC, ML, AB, LF, FP, EG, GCA, DC, MMi, SC, and LR collected the epidemiological, clinical data and were involved in data interpretation. SN and LG performed ELISA assay and processed statistical data. MG, SN, LG, and LR wrote the manuscript. All the authors critically revised the manuscript for intellectual content.

### Conflict of interest

The authors declare that they have no conflict of interest.

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
