## [Review Process File · EMBO Molecular Medicine]

Decreased serum level of sphingosine-1-phosphate: a novel predictor of clinical severity in COVID-19

Giovanni Marfia, Stefania Navone, Laura Guarnaccia, Rolando Campanella, Michele Mondoni, Marco Locatelli, Alessandra Barassi, Laura Fontana, Fabrizio Palumbo, Emanuele Garzia, Giuseppe Ciniglio Appiani, Davide Chiumello, Monica Miozzo, Stefano Centanni, and Laura Riboni

DOI: [10.15252/emmm.202013424](https://doi.org/10.15252/emmm.202013424)

Corresponding author: Giovanni Marfia (giovanni.marfia@policlinico.mi.it)

Review Timeline:

Submission Date:	15th Sep 20
Editorial Decision:	2nd Oct 20
Revision Received:	14th Oct 20
Editorial Decision:	27th Oct 20
Revision Received:	9th Nov 20
Accepted:	12th Nov 20

Editor: Zeljko Durdevic

Transaction Report:

2nd Oct 2020

Dear Dr. Marfia,

Thank you for the submission of your manuscript to EMBO Molecular Medicine. We have now received feedback from the three reviewers who agreed to evaluate your manuscript. As you will see from the reports below, the referees acknowledge the interest of the study and are overall supporting publication of your work pending appropriate revisions.

Addressing the reviewers' concerns in full will be necessary for further considering the manuscript in our journal, and acceptance of the manuscript will entail a second round of review. EMBO Molecular Medicine encourages a single round of revision only and therefore, acceptance or rejection of the manuscript will depend on the completeness of your responses included in the next, final version of the manuscript. For this reason, and to save you from any frustrations in the end, I would strongly advise against returning an incomplete revision.

I look forward to receiving your revised manuscript.

***** Reviewer's comments *****

Referee #1 (Comments on Novelty/Model System for Author):

This work is very timely and clinically significant. However, the methodology used to measure S1P is not optimal. LC/MS/MS methods are more accurate and the ELISA method employed is not as accurate.

Referee #1 (Remarks for Author):

This work is very timely and important. Sample collection and analysis seems to be done properly. The key weakness is that the ELISA method to measure S1P is not validated by a more rigorous method such as LC/MS/MS. I recommend that the authors validate some of the key findings in limited samples and show a concordancy between the two methods.

Referee #2 (Remarks for Author):

The paper by Marfia et al. reports that the serum level of sphingosine-1-phosphate could be a predictor of the severity of Covid-19. Overall, the data is well presented and discussed. The population is well selected and analyzed. Statistic analysis is appropriate. Few points are below.

1. ICU patients have low S1P level than no-ICU. Unclear how the authors claim that S1P is a "predictor" of ICU admission. If so, it should be lower in patients before being transferred/admitted to the ICU. A stratification of these patients is required.
2. ICU patients with low S1P have higher mortality (data provided in Figure 5). These data are difficult to interpret. It will be helpful to show a mortality rate of the ICU patients having less than 0.60 micromolar S1P and the mortality rate of the patients having more than 0.60 micromolar. This analysis should start from the day of ICU admission and not from the hospital admission, as the authors do not know the S1P level before the ICU admission in the same patient.
3. The lack of sequential analysis is a major limitation of this study (also acknowledged by the authors).
4. Figure 6 is highly complex. It should be modified to show the key findings.
5. That S1P level is low in Covid-19 patients is not novel. It has been already reported (see PMID:32610096), although that paper compares healthy versus Covid-19 patients and does not address severity.
6. Low level of S1P could simply be an epiphenomenon of host tissue damage and not directly linked to the damage caused by the SARS-CoV-2. In fact, low serum level of S1P has been suggested to predict mortality in patients with liver cirrhosis (PMID: 28334008); low serum level of S1P is associated with peripheral artery disease (PMID: 27973607), severity of septic patients (PMID: 31019718), in diabetic patients

Referee #3 (Remarks for Author):

1. These data are interesting and important.
2. Other health systems have been talking about this but no-one has laid out the data yet, so this is a first.
3. S1P receptor agonists (both fingolimod and ozanimod) are in clinical trials for Covid
4. The published data on cytokine storm in H1N1 2009 from Oldstone, Kawaoka and colleagues, shows that S1P receptor agonists protect from cytokine storm, while antagonists are deleterious. These data have been demonstrated on human plasmacytoid dendritic cells and the auto amplification loop for IFN α . Furthermore, IFNAR1 is downmodulated through s1PR1. These papers should perhaps be cited as supporting the mechanisms underlying predictive elements seen here.
5. I would expedite revision and publication

1.

S1PR1-mediated IFNAR1 degradation modulates plasmacytoid dendritic cell interferon- α autoamplification.

Teijaro JR, Studer S, Leaf N, Kiosses WB, Nguyen N, Matsuki K, Negishi H, Taniguchi T, Oldstone MB, Rosen H.

Proc Natl Acad Sci U S A. 2016 Feb 2;113(5):1351-6. doi: 10.1073/pnas.1525356113. Epub 2016 Jan 19.

PMID: 26787880 Free PMC article.

2.

Cytokine storm plays a direct role in the morbidity and mortality from influenza virus infection and is chemically treatable with a single sphingosine-1-phosphate agonist molecule.

Oldstone MB, Rosen H.

Curr Top Microbiol Immunol. 2014;378:129-47. doi: 10.1007/978-3-319-05879-5_6.

PMID: 24728596 Free PMC article. Review.

3.

Protection of ferrets from pulmonary injury due to H1N1 2009 influenza virus infection: immunopathology tractable by sphingosine-1-phosphate 1 receptor agonist therapy.

Teijaro JR, Walsh KB, Long JP, Tordoff KP, Stark GV, Einfeld AJ, Kawaoka Y, Rosen H, Oldstone MB. Virology. 2014 Mar;452-453:152-7. doi: 10.1016/j.virol.2014.01.003. Epub 2014 Jan 31.

PMID: 24606692 Free PMC article.

4.

Mapping the innate signaling cascade essential for cytokine storm during influenza virus infection.

Teijaro JR, Walsh KB, Rice S, Rosen H, Oldstone MB.

Proc Natl Acad Sci U S A. 2014 Mar 11;111(10):3799-804. doi: 10.1073/pnas.1400593111. Epub 2014 Feb 26.

PMID: 24572573 Free PMC article.

5.

Sphingosine-1-phosphate and its receptors: structure, signaling, and influence.

Rosen H, Stevens RC, Hanson M, Roberts E, Oldstone MB.

Annu Rev Biochem. 2013;82:637-62. doi: 10.1146/annurev-biochem-062411-130916. Epub 2013 Mar 18.

PMID: 23527695 Review.

6.

Endothelial cells are central orchestrators of cytokine amplification during influenza virus infection.

Teijaro JR, Walsh KB, Cahalan S, Fremgen DM, Roberts E, Scott F, Martinborough E, Peach R, Oldstone MB, Rosen H.

Cell. 2011 Sep 16;146(6):980-91. doi: 10.1016/j.cell.2011.08.015.

PMID: 21925319 Free PMC article.

7.

Suppression of cytokine storm with a sphingosine analog provides protection against pathogenic influenza virus.

Walsh KB, Teijaro JR, Wilker PR, Jatzek A, Fremgen DM, Das SC, Watanabe T, Hatta M, Shinya K, Suresh M, Kawaoka Y, Rosen H, Oldstone MB.

Proc Natl Acad Sci U S A. 2011 Jul 19;108(29):12018-23. doi: 10.1073/pnas.1107024108. Epub 2011 Jun 29.

PMID: 21715659 Free PMC article.

Authors' Response to Reviewers

Referee #1

This work is very timely and clinically significant. However, the methodology used to measure S1P is not optimal. LC/MS/MS methods are more accurate and the ELISA method employed is not as accurate.

Referee #1 (Remarks for Author):

This work is very timely and important. Sample collection and analysis seems to be done properly. The key weakness is that the ELISA method to measure S1P is not validated by a more rigorous method such as LC/MS/MS. I recommend that the authors validate some of the key findings in limited samples and show a concordancy between the two methods.

We thank the reviewer for recognizing the importance of our work and also for his/her recommendation.

We recognize that, as all methods, ELISA may have pitfalls. However, if this method is performed in proper conditions, it provides values consistent with those obtained by LC/MS/MS. In agreement, as we added to the text (pag. 7, lines 179 and 180), the levels of serum S1P in the control group were comparable with those reported by a recent study performed by LC/MS/MS on 174 healthy blood donors (Daum et al. 2020. Determinants of Serum- and Plasma Sphingosine-1-Phosphate Concentrations in a Healthy Study Group TH Open. 4(1):e12-e19, ref. n. 17). In addition, while LC/MS/MS is more rigorous, it should be noted that accurate quantification of serum S1P levels still poses many difficulties to LC/MS/MS technology, mainly due to variable effects of different biological matrices, as well as the lack of proper matrices free of analytes or samples with known concentrations of analytes (Tang et al. 2020. Validated LC-MS/MS method of sphingosine 1-phosphate quantification in human serum for evaluation of response to radiotherapy in lung cancer. Thoracic Cancer 11:1443-1452). Moreover, LC/MS/MS is significantly expensive and labor-intensive.

Despite we were unable to perform LC/MS/MS (the triple quadrupole mass spectrometer, required for S1P quantification, is not readily available in our laboratories), based on your recommendation, we performed new analyses to validate the serum level of S1P measured by ELISA. We verified our findings by enzymatic derivatization, which was reported to provide values that are very similar to those obtained by MS (Edsall L, Vann L., Milstien S., et al. Enzymatic measurement of sphingosine 1-phosphate, Methods Enzymol. 2000;312:9–16, ref n. 15). The results showed concordance between the two methods. The methodology of the enzymatic assay has been added to the text (pag. 5, lines 132-138).

Referee #2 (Remarks for Author):

The paper by Marfia et al. reports that the serum level of sphingosine-1-phosphate could be a predictor of the severity of Covid-19. Overall, the data is well presented and discussed. The population is well selected and analyzed. Statistic analysis is appropriate. Few points are below.

We thank the reviewer for his/her positive evaluation of the manuscript.

1. ICU patients have low S1P level than no-ICU. Unclear how the authors claim that S1P is a "predictor" of ICU admission. If so, it should be lower in patients before being transferred admitted to the ICU. A stratification of these patients is required.

We thank the reviewer for her/his queries, which allowed us to clarify that all blood samples were collected on the first day of hospitalization. This clarification has been added at page 4, line 114.

COV patients were then followed up and monitored for all their hospitalization time and when ICU admission was required they were classified into the ICU cohort, while the other patients who did not require ICU admission were categorized into the noICU group. A sentence was added to the text on pages 8,9, lines 207-209).

2. ICU patients with low S1P have higher mortality (data provided in Figure 5). These data are difficult to interpret. It will be helpful to show a mortality rate of the ICU patients having less than 0.60 micromolar S1P and the mortality rate of the patients having more than 0.60 micromolar. This analysis should start from the day of ICU admission and not from the hospital admission, as the authors do not know the S1P level before the ICU admission in the same patient.

Thank you for your suggestion, which allowed us to strengthen the relationship between S1P and patient mortality. As proposed, we stratified our COV population into patients with low ($< 0.60 \mu\text{M}$) and high ($\geq 0.60 \mu\text{M}$) levels of S1P. The results are reported on page 10, lines 242-245.

3. The lack of sequential analysis is a major limitation of this study (also acknowledged by the authors).

As we acknowledged in the discussion, a limitation of this study is the lack of sequential analyses. When we started this investigation, no data were available on S1P in COVID-19, and we concentrated on S1P levels at the time of admission. We recently started evaluating S1P at different time points in the same, new patients, and this will hopefully provide new insights, but it requires additional time.

4. Figure 6 is highly complex. It should be modified to show the key findings.

We simplified Figure 6, focusing on the main findings, Accordingly, the figure legend was also modified (page 13, lines 272-281).

5. That S1P level is low in Covid-19 patients is not novel. It has been already reported (see PMID:32610096), although that paper compares healthy versus Covid-19 patients and does not address severity.

We thank you for your comment and recent reference indication. The findings of this very recent paper were inserted in the first paragraph of the Discussion session (pag. 11, lines 254-257). As briefly commented in the text, we should note that the paper you cited did not reported S1P in molar concentrations, but as Intensities, preventing the comparison of data across independent studies. Notwithstanding our study appears consistent with this study, it provides novelty in addressing severity and mortality (as you acknowledge).

6. Low level of S1P could simply be an epiphenomenon of host tissue damage and not directly linked to the damage caused by the SARS-CoV-2. In fact, low serum level of S1P has been suggested to predict mortality in patients with liver cirrhosis (PMID: 28334008); low serum level of S1P is associated with peripheral artery disease (PMID: 27973607), severity of septic patients (PMID: 31019718), in diabetic patients.

Thank you for this comment. We agree with the Referee that we cannot exclude that the decreased levels of S1P could be an epiphenomenon of host tissue damage. However, taking into account the pleiotropic effects of S1P on multiple organs, and that S1P plays crucial roles in endothelial dysfunction, as well as in immunity, we think reasonable the possibility that low S1P levels contributes to COVID-19 complications. The evidence that low serum level of S1P predicts mortality in patients with the different diseases (you reported) appear to support a role of S1P in the progression of different diseases, as endothelial and immune alterations which are

common to these diseases, as well as COVID-19. According to reviewer's suggestions, we added a new piece of text, mentioning the studies recommended by the reviewer in the discussion section (pages 15,16, lines 348-349, 352-356, references n. 14, 31, 32).

Referee #3 (Remarks for Author):

1. These data are interesting and important.

Thank you for your comment.

2. Other health systems have been talking about this but no-one has laid out the data yet, so this is a first.

Thank you for recognizing the novelty of our study.

3. S1P receptor agonists (both fingolimod and ozanimod) are in clinical trials for Covid.

4. The published data on cytokine storm in H1N1 2009 from Oldstone, Kawaoka and colleagues, shows that S1P receptor agonists protect from cytokine storm, while antagonists are deleterious. These data have been demonstrated on human plasmacytoid dendritic cells and the auto amplification loop for IFN α . Furthermore, IFNAR1 is downmodulated through s1PR1. These papers should perhaps be cited as supporting the mechanisms underlying predictive elements seen here.

Thank you for the suggestion. According to it, we enriched the discussion by reporting about the controversial effect of the modulation of S1P receptor by Fingolimod in viral infection (page 16, lines 368-371, references n. 35,36).

5. I would expedite revision and publication

Thank you. We appreciated your positive evaluation.

27th Oct 2020

Dear Dr. Marfia,

Thank you for the submission of your revised manuscript to EMBO Molecular Medicine. I am pleased to inform you that we will be able to accept your manuscript pending the following final amendments:

Please implement all adjustments suggested by the referee #1. No additional experiments are required.

***** Reviewer's comments *****

Referee #1 (Remarks for Author):

The authors have addressed my comments in part by using a classical enzymatic method to assess S1P, in which S1P is extracted, phosphorylated and estimated by radioactive phosphate incorporation and TLC analysis. The results, which are not shown, apparently are in concordance with the ELISA methods. While this may be so, the authors should acknowledge that this classic method is prone to inaccuracies, such as inefficient lipid extraction at multiple steps, misidentification of other lipids that run at the same R_f value as S1P on the TLC plate, etc. The state of the art technique that is the most accurate to quantify S1P in biological samples is one step extraction followed by LC/MS/MS. Acknowledging this is important for the scientific literature. The authors should at the very least acknowledge this in the limitations of their study. Ideally, they should send a few of their samples (high and low S1P) to a LC/MS/MS core facilities or metabolomics centers and show correlation between the two. This would strengthen the study tremendously.

Referee #2 (Comments on Novelty/Model System for Author):

Highly relevant study that may help in the stratification of the severity of the COVID 19 disease.

Referee #2 (Remarks for Author):

The authors responded to my comments in a satisfactory manner

Referee #3 (Remarks for Author):

The revisions have improved the readability and clarity of the manuscript. I have no further suggestions

The authors performed the requested changes.

The authors performed the requested changes.

Corresponding Author Name: Giovanni Marfia

Manuscript Number: EMM-2020-13424-V2